# Optimization of Jinhua Ham Classification Method Based on Volatile Flavor Substances and Determination of Key Odor Biomarkers

**DOI:** 10.3390/molecules27207087

**Published:** 2022-10-20

**Authors:** Ying Xu, Mengzhu Shui, Da Chen, Xiaozhong Ma, Tao Feng

**Affiliations:** 1School of Perfume and Aroma Technology, Shanghai Institute of Technology, Shanghai 201400, China; 2Department of Animal, Veterinary and Food Sciences, University of Idaho, 875 Perimeter Drive, Moscow, ID 83844, USA; 3Jinzi Ham Co., Ltd., No. 1000, Jinfan Street, Industrial Park, Jinhua 321016, China

**Keywords:** HS-SPME, GC–MS, Jinhua ham, flavor, OAV

## Abstract

Jinhua ham is a traditional cured meat food in China. For a long time, its grade has mainly been evaluated by the human nose through the three-sticks method, which is highly subjective and is not conducive to establishing evaluation standards through odor markers. In this paper, we analyzed the well-graded Grade I–III hams provided by Jinzi Ham Co., Ltd. (Jinhua, China). Firstly, we used different extraction fibers, extraction temperatures, and extraction time to determine the optimal conditions for headspace solid-phase microextraction (HS-SPME). Then, the aroma components of Jinhua ham were analyzed by headspace solid-phase microextraction combined with gas chromatography–mass spectrometry (GC–MS), and OAV was calculated to screen the key aroma volatiles of three kinds of Jinhua ham. It was found that a total of 56 components were detected in the three types of ham. Among them, there are 21 kinds of key aroma volatiles. Aldehydes, alcohols, and acids are the three main components of Jinhua ham, and the content of aldehydes gradually decreases from Grade I to Grade III ham. The content of acids gradually increased, and we speculated that the increase in acid content was caused by the proliferation of microorganisms in Grade III ham. The key flavor volatiles in Grade I hams was hexanal and 2-methylbutanal. Grade I hams had a strong meat aroma, pleasant fatty, and roasted aroma without any off-flavors. In Grade II ham, the characteristic volatiles (E,E)-2,4-decadienal and ethyl isovalerate were detected. These two volatiles contribute greatly to the flavor of Grade II ham, which makes the flavor of Grade II ham have a special fruity aroma. They also may be prone to sourness and affect the flavor of the ham. Volatiles with low threshold values, such as pyrazines, furans, and sulfur-containing compounds, were relatively high in Grade III hams. This may also contribute to the poorer flavor quality of Grade III hams. This experiment provided a reliable test method and evaluation basis for the rating of Jinhua ham. These results have positive implications for the establishment of odor markers-based grading criteria.

## 1. Introduction

Ham is recognized as one of the three most nutritious fermented foods in the world [1]. The world’s most famous dry-cured hams, such as Kumpiak podlaski, Parma, Kraški pršut, and Jinhua dry-cured hams, differ in terms of the raw materials and the way they are processed. Kumpiak podlaski, Parma, and Kraški pršut dry-cured hams are generally made from large noble breeds such as Polish Landrace or Polish Large White [2,3]. Jinhua Ham, produced locally in Jinhua City, is made from the hind legs of the traditional Jinhua pig breed. This breed of pig has black skin and black hair on the neck and rump and white skin and white hair on the chest, belly, and limbs. Production involves salting, washing, sun-drying, shaping, ripening, and post-ripening of pig legs [4,5]. This is mainly attributed to the fermentation process during production. It not only degrades proteins to peptides for improved absorption [6,7] but also decomposes fat to be free fatty acids and small aromatic compounds, which endows typical ham flavor.

In fermented meat products, volatiles are commonly generated by the oxidation of unsaturated fatty acids followed by reactions with amino acids and peptides. The main volatiles include hydrocarbons, aldehydes, alcohols, acids, ketones, lactones, and other compounds such as benzene derivatives, amines, and amino compounds [8,9,10]. Currently, the main methods for extracting ham volatiles are simultaneous distillation extraction, dynamic headspace technique, solid-phase microextraction, supercritical fluid extraction, and thermal sorption, combined with gas chromatography–mass spectrometry (GC–MS) for separation and identification [11,12,13,14,15]. Headspace solid-phase microextraction (HS-SPME) is a simple and fast approach to extract volatile compounds without the usage of organic solvents, which has been widely adopted nowadays to screen out the flavor differences among different samples. However, volatiles are affected by extraction conditions, such as differences in the coating of the extracted fibers. Extracted fibers can be broadly classified into three different types, depending on the coating. The first type is polar coating, such as polyacrylate (PA), which is suitable for adsorbing polar compounds and has been used in the analysis and detection of pesticides, fatty acids, food flavors, and phenols [16]. The second type is non-polar coatings, such as polydimethylsiloxane (PDMS), which is suitable for the absorption of non-polar and weakly polar compounds. It has been used for the analysis and detection of organic chlorine, organic phosphorus, organic nitrogen pesticides, drugs, narcotics, food aroma, and caffeine [17]. The third type is moderately polar hybrid coatings, including Carboxen/Polydimethylsiloxane (CAR/PDMS) and Divinylbenzene/Carboxen/Polydimethylsiloxane (DVB/CAR/PDMS) [18,19]. Roberts found that the Carboxen/Polydimethylsiloxane (CAR/PDMS) was the most effective for small molecules and acids [20]. The DVB/CAR/PDMS coating shows better extraction efficiency for aromatic homologs [18].

The traditional three-sticks method is used by an experienced ham judge to determine the aroma of the sticks. The judge inserts three sticks into the ham, and the ham with a strong meat aroma and no off-flavor on any of the three sticks is judged to be Grade Ⅰham. This method is highly influenced by human factors. In the present project, we used HS-SPME-GC–MS to detect and evaluate the quality of Jinhua ham based on its volatiles. Different extracting fibers (85 μmPA, 100 μm PDMS, 75 μm CAR/PDMS, aand 50/30 μm DVB/CAR/PDMS), extracting time, and temperatures were optimized to acquire the most suitable extraction conditions for Jinhua ham volatiles. This experiment provided a reliable test method and evaluation basis for the rating of Jinhua ham.

## 2. Results and Discussion

### 2.1. Optimal Conditions for SPME Extraction

When fixing other conditions but changing different extraction fibers, the detection results are shown in Figure 1. The adsorption capacity of different extraction fiber fibers for Jinhua ham flavor substances varied. The number of compounds and peak areas detected by the composite coating was twice as high as those detected by the single polar coating and the non-polar coating. In comparison, the 75 μm CAR/PDMS extraction tip had better performance on sensitivity, adsorption capacity, and chromatographic response value, so it was chosen as the extraction fiber tip for detecting the flavors of Jinhua ham. In Huang’s study, the CAR/PDMS fiber was selected for the SPME Arrow method for subsequent analysis of furan and its derivatives in commercial food products by GC–MS/MS [21].

The extraction time also had a large effect on the number of adsorbed flavors. Due to the distinct equilibration time among different samples, the optimal extraction time also varied; the detection results are shown in Figure 2. We compared 30 min, 40 min, and 50 min and found with the increase in extraction time, the total peak area gradually increased, indicating a higher concentration. However, the number of detected volatiles began to decrease after 40 min. Excessive extraction time may destroy ham volatiles. Meanwhile, as the extraction time increased, the extracted fibers tended to saturate, which could affect the effect of extractive adsorption and can even lead to desorption problems. Therefore, 40 min was chosen as the optimal extraction time.

Extraction temperature has a dual effect on the samples. On the one hand, a higher temperature accelerates the movement of odor molecules in the sample and increases the vapor pressure, which is beneficial to extraction, especially for headspace solid-phase microextraction. On the other hand, a higher temperature may reduce the adsorption capacity of the extractor [22]. It can be seen from Figure 3 that when the extraction temperature was low, the number and types of volatiles in the headspace bottle were limited. With the increase in temperature, the total area of the peaks was increased, but the number of identified volatiles decreased. Moreover, if the temperature is too high, it may destroy the original volatiles of the ham and generate some other substances, which will affect the original flavor of Jinhua ham, so 60 °C is the best temperature for extraction.

The value of the total peak area in the above figure is also multiplied by 10^9^ times, the same as below.

### 2.2. GC–MS Analysis of Different Grades of Jinhua Ham

The spectra obtained under the optimal extraction conditions were analyzed qualitatively and quantitatively with data analysis software, and the Mylab and Nist standard atlas libraries were used for data retrieval. The identification results were as follows.

It can be seen from Table 1 that a total of 56 volatiles were detected in the three grades of ham, of which 32, 33, and 42 volatiles were detected in the hams of Grades I–III, respectively. There were 17 aldehyde volatiles detected, followed by alcohol volatiles and acid volatiles, with 10 and 9, respectively. Aldehyde volatiles were the dominant component of Jinhua ham. From Grade I ham to Grade III ham, the content of aldehydes decreased gradually. This conclusion was consistent with the studies of Liu and Huan [5,23]. As can be seen from Table 1, the aldehyde volatiles hexanal, valeraldehyde, 2-methylbutanal, isovaleraldehyde, heptanal, benzaldehyde, and phenylacetaldehyde are all represented in a large proportion of the three grades of ham. Among them, isovaleraldehyde was one of the typical aromas of Jinhua hams [5]. The study by Liu et al. showed that hexanal, heptanal, benzaldehyde, and phenylacetaldehyde also play important roles in lipid oxidation and protein degradation in dry-cured ham. Hexanal was present in high amounts in all three types of ham and was the main volatile compound in ham. Hexanal has a distinct aldehyde and fatty odor(Reference database: TGSC Information System). The total alcohol volatile content of Grade I hams was more prominent compared to Grade II and III hams, and the types and contents of alcohols in Grade II and III hams were similar, with no major differences. During the fermentation process of ham, due to the Maillard reaction of aldehydes and ketones, acid volatiles were produced, which was one of the unique flavor sources of Jinhua ham. The acid volatiles increased step by step from Grade I to Grade III hams in terms of acid content. Smaller molecular weight acids such as propionic acid, butyric acid, valeric acid, and capric acid possess a cheese aroma. The acids (C_1_–C_6_) imparted a sour taste but can be neutralized by alkaline compounds [24]. Alkane volatiles and ketone volatiles were not significantly different in the three types of hams. In Grade I-III hams, the content of both oxygenated compounds and multiple pyrazine volatiles showed a gradual increase. 2-Pentylfuran is an oxidation product of linoleic acid, which also has a ham-like flavor. Five pyrazine volatiles were detected in Grade III ham, and the content was higher than that in Grade I and Grade II ham. However, due to the high threshold of pyrazine substances, it contributed negligibly to the flavor of the ham. The sulfur-containing volatiles dimethyl disulfide and dimethyl trisulfide were detected in Grade III hams, and no sulfur-containing volatiles were detected in Grades I and II hams. The low threshold of sulfur-containing volatiles and their large content in Grade III hams may have some negative effects on the flavor of Grade III hams.

### 2.3. Key Volatiles in Jinhua Ham

By querying the threshold of volatiles and calculating OAV, we screened volatiles with an OAV value ≥ 1. The results are shown in Table 2. A total of 21 key volatiles were identified in the three hams, and 15, 15, and 16 volatiles were detected in Grade I, Grade II, and Grade III hams, respectively. Among the key volatiles, 14 aldehyde volatiles were identified. Because the threshold values of aldehyde volatiles are generally low, they have a greater impact on the flavor of ham. Fat oxidation and decomposition was very important chemical reaction in the process of ham curing and fermentation, and the free fatty acids produced by the decomposition had an important contribution to the flavor of ham. The products of fatty oxidation mainly include linear aldehydes, alkenes, ketones, and alcohols. The content and OAV value of hexanal in the three kinds of hams were very high and were the key volatile of hams. Hexanal was mainly produced by the degradation of unsaturated fatty acids. Isovaleraldehyde and 2-methylbutanal have fruity, green, and nutty aromas [25]. (E,E)-2,4-decadienal and 2,4-nonadienal both have fried and oily aromas(TGSC Information System). (E,E)-2,4-decadienal and 2,4-nonadienal were produced by Strecker degradation of amino acids such as isoleucine and leucine and played an important role in the flavor of ham.

Most alcohols have high thresholds and contribute little to flavor, but the threshold of oct-1-en-3-ol was low (0.0015 mg/kg), with mushroom-like, green, and greasy odor(TGSC Information System). The content of alkanes, alkenes, ketones, and pyrazines detected in the ham was low and had no significant effect on the flavor of the ham. Among the esters, short-chain esters have a pleasant fruity taste, while long-chain esters have a greasy taste [26]. The esters were derived from the esterification of carboxylic acids and alcohols. Ethyl isovalerate was only detected in Grade II ham, and its threshold was low, which had a greater impact on the flavor of Grade II ham. Ethyl isovalerate had a sweet and fruity aroma. Both dimethyl disulfide and dimethyl trisulfide were sulfur-containing compounds detected only in Grade III hams, with an unpleasant odor similar to sulfur and onion [27]. The main reason might be that in the process of ham processing, the fermentation temperature was higher in the later stage, and sulfur-containing proteins and amino acids were degraded to produce sulfur-containing compounds. The metabolism of microorganisms was also responsible for the production of sulfur-containing compounds [5].

In summary, as shown in Figure 4, it was shown that Grade I hams had the most abundant aldehyde species, and hexanal and 2-methylbutanal contributed the most to the flavor of Grade I ham. The Grade Ⅰ ham had a strong meaty and roasted aroma, as well as a pleasant fatty aroma. In Grade II ham, (E,E)-2,4-decadienal and ethyl isovalerate had the greatest influence on its flavor. In addition to the pleasant meaty flavor, the fruitiness was also more pronounced in the overall flavor of the Grade II ham. Excess fruitiness may adversely affect the flavor of Grade II ham. More off-flavor compounds, such as 2-amylfuran, dimethyl disulfide, and dimethyl trisulfide, were significantly detected in Grade III hams, which negatively impacted the overall flavor and contributed to the poorer aroma variety.

## 3. Materials and Methods

### 3.1. Materials and Reagents

Jinhua ham cubes (Grade I, II, and III) were provided by China Jinzi Ham Co., Ltd. Jinhua ham was selected from the hind legs of a traditional local breed of pig as the raw material. The raw material was pre-treated and then cured at 8 °C for about 35 days. After curing, the ham was immersed in water at 10 °C and brushed. The ham was then hung up to dry until the skin was shiny yellow, and the meat was spread with oil. Finally, the ham was fermented, which usually took about 5 months. Fermented and trimmed hams were stacked individually on a wooden bed according to the size and dryness of the hams and turned over every 5–7 days.

We peeled the skin of the graded Jinhua ham and then removed the fat and bones. Afterward, the ham was chopped evenly, packed in an airtight bag, and stored in the refrigerator. The ham was taken out from the refrigerator and left at room temperature for 30 min before usage.

O-dichlorobenzene (99%) and C8-C30 series alkanes (analytical grade) were purchased from Sinopharm Chemical Reagent Co., Ltd., Shanghai, China, and Sigma Aldrich (Burlington, MA, USA), respectively.

### 3.2. Experimental Method

#### 3.2.1. Optimization of Single Factor of Extracting Conditions

We made minor adjustments according to the experimental conditions of Sánchez-Peña. [28] Chopped ham (4.5 g) was weighed and placed into 15 mL glass vials tightly capped with a PTFE septum, and 5 μL of *o*-dichlorobenzene was added as an internal standard. Condition optimization was conducted from four different types of extractors (85 μm PA is polar coated, 100 μm PDMS is non-polar coated, 75 μm CAR/PDMS, and 50/30 μm DVB/CAR/PDMS is a moderately polar hybrid coating (Supelco, St. Louis, MO, USA)), three different extraction time (30 min, 40 min, and 50 min) and three different extraction temperatures (50 °C, 60 °C, 70 °C).

#### 3.2.2. Gas Chromatography–Mass Spectrometry (GC–MS)

The gas chromatograph–mass spectrometer (Agilent 7890A-5975C, Agilent Technologies, Santa Clara, CA, USA) was used for the identification of ham volatiles. The electron ionization energy of MSD was 70 eV. The temperature of the inlet was set at 250 °C, and the temperature of the detector was set at 280 °C. The ion source temperature was set at 230 °C. Volatiles were separated by using HP-INNOWAX analytical fused silica capillary column (60 m × 0.25 mm × 0.25 μm, Agilent Technologies, Santa Clara, CA, USA). Helium was used as the carrier gas at a flow rate of 0.8 mL/min. The quadruple mass filter and transfer line temperature were operated at 150 °C and 280 °C, respectively. The GC oven temperature was initially 50 °C, then ramped up to 60 °C at 5 °C/min, and continually for ten minutes, then ramped up to 100 °C at 8 °C/min, and continually for 8 min, then ramped up to 200 °C at 15 °C/min, and continually for 10 min.

### 3.3. Qualitative and Semi-Quantitative

For qualitative analysis, the original data were matched with the NIST database, and compounds with a match degree (SI) > 800 were analyzed. By calculating the retention index and comparing it with the database (https://webbook.nist.gov/chemistry/cas-ser/ (accessed on 2 August 2022)), the qualitative volatiles were further screened. The calculation Formula (1) is as follows:RI = 100 × *n* + 100(t_a_ − t*_n_*)/(t_*n*+1_ − t*_n_*)(1)

t_a_, retention time of chromatographic peak a;

t*_n_*, t_*n*+1_, retention time of *n*-alkanes C*_n_* and C_*n*+1_.

For semi-quantitative analysis: the content of the compound was calculated according to the internal standard [29] using the formulation below:C_a_ = C_i_ × (A_a_/A_i_)(2)

C_a_, the mass concentration of the volatile (μg/kg);

C_i_, the mass concentration of the internal standard (μg/kg);

A_a_, the chromatographic peak area of volatiles;

A_i_, the chromatographic peak area of the internal standard.

### 3.4. Odor Activity Value (OAV)

The aroma contribution of volatiles was evaluated by the odor activity value OAV to determine the key flavor volatiles [30]. The calculation formula is shown in Formula (3):OAV = C_x_/T_x_(3)

C_x_, the mass concentration of the compound (μg/kg);

T_x_, sensory threshold (μg/kg) of the compound.

### 3.5. Statistical Analysis

The spectra obtained from the extraction analysis were analyzed qualitatively and quantitatively using the Mylab and Nist.11 standard atlas libraries, and the data were retrieved using Mylab and Nist.11 standard spectral libraries. Microsoft Excel 2016 (Microsoft, Redmond, WA, USA) was used to statistically filter the data. Origin2022 (Origin Lab, Northampton, MA, USA) was used to plot the graphs. Each experiment was repeated three times.

## 4. Conclusions

In this experiment, the optimal conditions for HS-SPME extraction of ham were acquired by verifying the effects of different extraction fibers, extraction temperature, and extraction time on the extraction of ham volatiles: the extraction was carried out in a water bath at 60 °C for 40 min using a 75 μm CAR/PDMS extraction fiber, and the sample was injected and resolved at 250 °C for 5 min. The study showed that the content of the first major flavor component, aldehyde, gradually decreased from Grade I to Grade III hams, whereas the content of the third major flavor component, acid, tended to increase more significantly. Grade 1 ham had a low ester content. Grade III hams contained a greater variety and content of esters than the other two. The content of hexanal in the three kinds of ham was the highest, and it also had a great influence on the flavor of the ham. The thresholds of nitrogen-containing and sulfur-containing volatiles were lower, which contributed more to the flavor of the ham.

The types of aldehydes in Grade I ham were relatively abundant, and almost no other odorous volatiles were detected. The Grade Ⅰ ham had a strong meaty and roasted aroma, as well as a pleasant fatty aroma. Hexanal and 2-methylbutanal contributed the most to the flavor of the Grade Ⅰ ham. Of these, 2-methylbutanal had significantly higher OAV values than the other two types of hams in the Grade Ⅰ ham. The OAV values of (E,E)-2,4-decadienal and ethyl isovalerate were large in Grade II ham, and these two volatiles were only detected in Grade II ham, which contributed significantly to its flavor. Both of these volatiles had a fruity aroma, so the fruitiness would be more prominent in the aroma of a Grade II ham than the other two. Much higher levels of pyrazine volatiles were detected in the tertiary ham than in the other two hams, while dimethyl disulfide and dimethyl trisulfide were only detected in the tertiary ham. Pyrazine volatiles had a roasted and roasted meat odor. Sulfide has a characteristic sulfur-like odor. These compounds impart a pleasant fragrance in low concentrations and a disgusting taste in high concentrations. The increased content of these substances in Grade III ham is also the reason for its poor aroma quality. This research provides a more reliable detection method and theoretical basis for ham rating in the future.

## Figures and Tables

**Figure 1 molecules-27-07087-f001:**
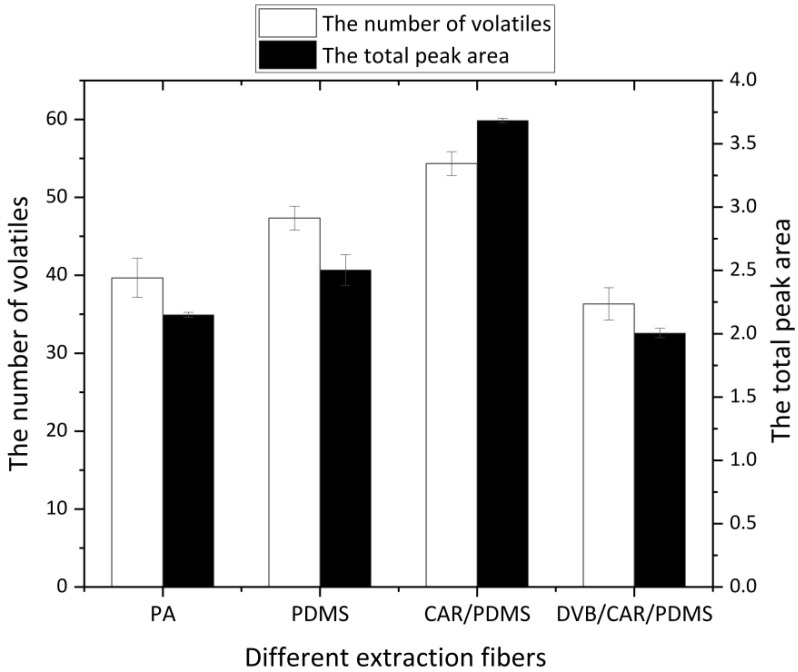
Comparison of extraction results of different extraction fibers.

**Figure 2 molecules-27-07087-f002:**
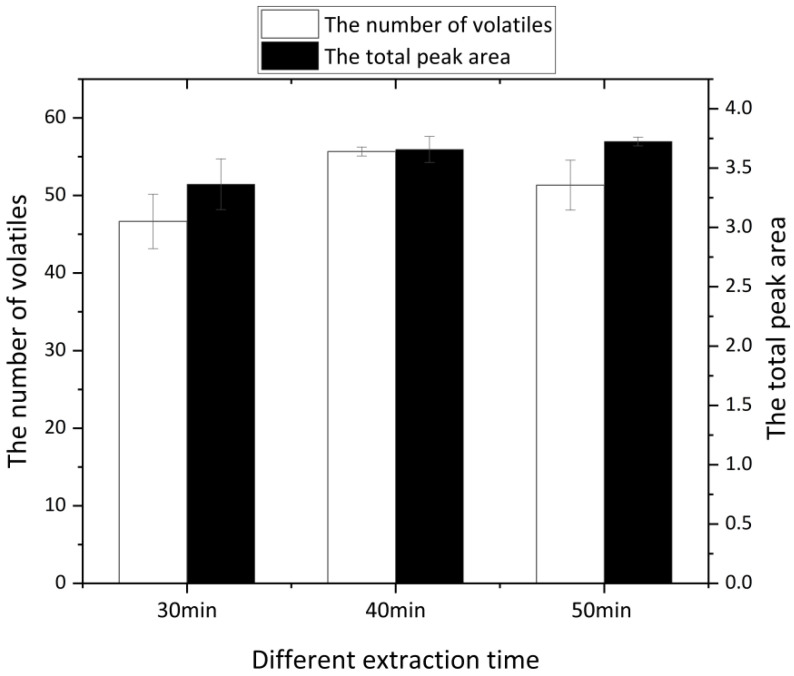
Comparison of extraction results for different extraction times.

**Figure 3 molecules-27-07087-f003:**
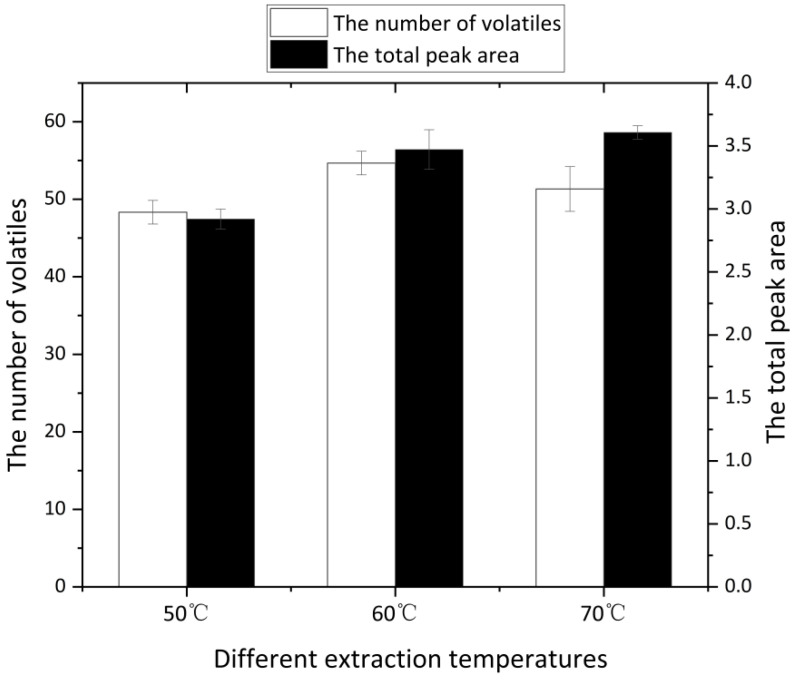
Comparison of extraction results at different extraction temperatures.

**Figure 4 molecules-27-07087-f004:**
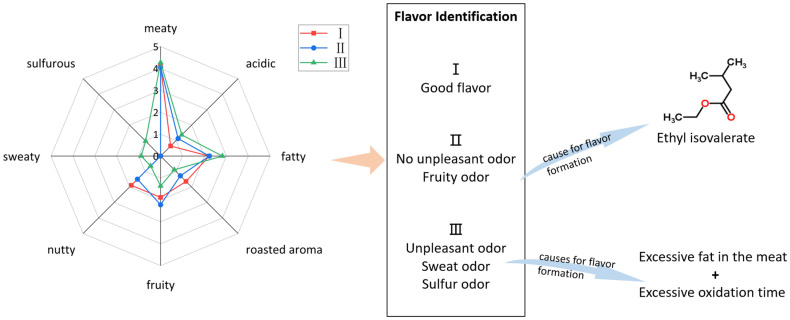
Flavor grade identification chart of Jinhua ham.

**Table 1 molecules-27-07087-t001:** Analysis results of flavor volatiles of Jinhua ham from Grades I to III.

NO.	Compounds	CAS	RI		Concentration (μg/kg)		
			Reference	Calculate	Grade Ⅰ	Grade II	Grade III
Aldehydes							
1	hexanal	66-25-1	1084	1076	989	889.7	919.8
2	heptenal	18829-55-5	1332	1340	6.1	2.6	16.1
3	benzaldehyde	100-52-7	1520	1519	14.5	13.5	18.3
4	octanal	124-13-0	1291	1287	9.8	7.4	- ^a^
5	(E,E)-2,4-Heptadienal	4313-03-5	1497	1503	-	-	1.3
6	2,4-decadienal	2363-88-4	1762	1763	0.5	-	-
7	phenylacetaldehyde	122-78-1	1650	1639	8.4	1.2	1.2
8	2-octenal	2363-89-5	1445	1438	1.8	-	6.1
9	nonanal	124-19-6	1382	1367	14.1	20.7	-
10	(E)-2-nonenal	18829-56-6	1551	1587	1.2	-	36.5
11	decanal	112-31-2	1485	1466	0.7	3.9	20.5
12	heptanal	111-71-7	1186	1186	21.3	29.6	14.5
13	(E,E)-2,4-decadienal	25152-84-5	1819	1810	-	10.3	-
14	2,4-nonadienal	6750-03-4	1710	1702	-	-	1.8
15	isovaleraldehyde	590-86-3	936	954	19.7	27.7	42.6
16	2-methylbutanal	96-17-3	926	950	155	58.5	4.4
17	valeraldehyde	110-62-3	1013	1029	177	89.4	44.5
Total					1419.1	1154.5	1127.6
Alcohols							
18	1-octanol	111-87-5	1559	1598	0.3	-	-
19	1-penten-3-ol	616-25-1	1157	1177	41.9	-	-
20	3-methyl-1-butanol	123-51-3	1208	1196	33.6	30.3	5.1
21	2-methylbutan-1-ol	137-32-6	1197	1160	11.6	13.2	6.6
22	pentanol	71-41-0	1210	1235	127.1	33	35
23	1-hexanol	111-27-3	1384	1407	5.1	-	4.9
24	oct-1-en-3-ol	3391-86-4	1456	1452	22	2	15.6
25	phenylethyl alcohol	60-12-8	1931	1979	0.5	-	-
26	heptan-1-ol	111-70-6	1456	1473	-	4.1	3.8
27	2-cyclohexenol	822-67-3	1471	1468	-	-	1.7
Total					242.1	82.6	72.7
Esters							
28	ethyl caprate	110-38-3	1624	1659	-	3	33.2
29	γ-butyrolactone	96-48-0	1595	1588	1.5	-	-
30	ethyl isovalerate	108-64-5	1067	1066	-	2.7	-
Total					1.5	5.7	33.2
Acids							
31	acetic acid	64-19-7	1465	1449	54.3	106.4	109.4
32	3-methylbutanoic acid	503-74-2	1624	1589	49.9	84.8	83.8
33	2-methylbutyric acid	116-53-0	1674	1696	12	36.2	63.8
34	1-hexanoic acid	142-62-1	1849	1822	31	4.1	28.1
35	propionic acid	79-09-4	1486	1502	-	23.6	23
36	isobutyric acid	79-31-2	1584	1608	-	29.2	4.4
37	3-methylvaleric acid	105-43-1	1762	1766	-	6.1	-
38	butyric acid	107-92-6	1628	1632	-	-	33.8
39	valeric acid	109-52-4	1730	1735	-	-	23.8
Total					147.2	290.4	370.1
Alkyl hydrocarbons							
40	limonene	138-86-3	1200	1244	2	4.8	5.6
41	ethylcyclopentene	2146-38-5	891	905	-	3.8	-
Total					2	8.6	5.6
Ketones							
42	3-octanone	106-68-3	1241	1256	8.6	-	-
43	2-heptanone	110-43-0	1184	1180	1.3	-	6.6
44	amyl ketone	927-49-1	1527	1562	-	10.2	0
45	3-hexanone	589-38-8	1068	1093	-	-	6.9
46	octane-2,5-dione	3214-41-3	1319	1301	-	-	8.8
47	3,5-octadienone,3,5-octadien-2-one	38284-27-4	1569	1573	-	-	1.4
48	2-decanone	693-54-9	1480	1500	-	-	0.5
Total					9.9	10.2	24.2
Oxygen-containing heterocyclic compounds							
49	2-amylfuran	3777-69-3	1231	1245	0.3	0.7	22.9
Nitrogen-containing heterocyclic compounds							
50	2,6-dimethylpyrazine	108-50-9	1327	1334	0.6	0.7	2.7
51	2,3,5-trimethylpyrazine	14667-55-1	1413	1418	-	0.5	19.7
52	tetramethylpyrazine	1124-11-4	1466	1481	-	-	16.7
53	2,5-dimethylpyrazine	123-32-0	1332	1337	-	1.4	-
54	2-ethyl-6-methylpyrazine	13925-03-6	1363	1376	-	-	1.3
Total					0.6	2.6	40.4
Sulfur compounds							
55	dimethyl disulfide	624-92-0	1078	1101	-	-	14.2
56	dimethyl trisulfide	3658-80-8	1400	1413	-	-	4.6
Total					-	-	18.8

^a^: ‘-’ means non-detectable.

**Table 2 molecules-27-07087-t002:** Key volatiles with OAV ≥ 1 in ham.

NO.	Compounds	Threshold (mg/kg) ^a^	OAV			Odor ^b^
			I	II	III	
1	hexanal	0.005	197.8	177.9	184.0	fresh green fatty aldehydic grass leafy fruity sweaty
2	octanal	0.000587	16.70	12.61	- ^c^	aldehydic waxy citrus orange peel green herbal fresh fatty
3	2,4-decadienal	0.0003	1.667	-	-	orange sweet fresh citrus fatty green
4	phenylacetaldehyde	0.0063	1.333	0.190	0.190	green sweet floral hyacinth clover honey cocoa
5	2-octenal	0.0002	9.000	-	30.50	fatty green herbal
6	nonanal	0.0011	12.82	18.82	-	waxy aldehydic rose fresh orris orange peel fatty peely
7	(E)-2-nonenal	0.00019	6.316	-	192.1	fatty green cucumber aldehydic citrus
8	decanal	0.003	0.233	1.300	6.833	sweet aldehydic waxy orange peel citrus floral
9	heptanal	0.0028	7.607	10.57	5.179	fresh aldehydic fatty green herbal wine-lee ozone
10	(E,E)-2,4-decadienal	0.000027	-	381.5	-	oily cucumber melon citrus pumpkin nut meat
11	2,4-nonadienal	0.00005	-	-	36	fatty green cucumber
12	isovaleraldehyde	0.0011	17.91	25.18	38.73	ethereal aldehydic chocolate peach fatty
13	2-methylbutanal	0.0011	140.9	53.18	4.000	musty cocoa phenolic coffee nutty malty fermented fatty alcoholic
14	valeraldehyde	0.012	14.75	7.450	3.708	fermented bready fruity nutty berry
15	3-methyl-1-butanol	0.004	8.400	7.575	1.275	fusel oil alcoholic whiskey fruity banana
16	oct-1-en-3-ol	0.0015	14.67	1.333	10.40	mushroom earthy green oily fungal raw chicken
17	ethyl isovalerate	0.00001	-	270.0	-	fruity sweet apple pineapple tutti frutti
18	isobutyric acid	0.0054	-	5.407	0.815	acidic sour cheese dairy buttery rancid
19	2-amylfuran	0.0058	0.052	0.121	3.948	fruity green earthy beany vegetable metallic
20	dimethyl disulfide	0.0011	-	-	12.91	sulfurous vegetable cabbage onion
21	dimethyl trisulfide	0.0001	-	-	46.00	sulfurous cooked onion savory meaty

^a^: Odor thresholds were taken from Van Gemert (2003). ^b^: Query website for the aroma of volatiles: http://www.thegoodscentscompany.com/ (accessed on 10 August 2022). ^c^:‘-’ means non-detectable.

## Data Availability

The data used to support the findings of this study are included within the article.

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
