# Peer review of "Optimization of Jinhua Ham Classification Method Based on Volatile Flavor Substances and Determination of Key Odor Biomarkers"

_molecules, 2022, doi:10.3390/molecules27207087_

Round 1

Reviewer 1 Report

The manuscript (molecules-1970726) determined the optimal conditions for headspace solid-phase microextraction (HS-SPME) of ham and screened the key aroma volatiles of three kinds of Jinhua ham. These results have positive implications for the establishment of odor markers-based grading criteria for Jinhua ham. This work is performed well and the story is clear. The following points are suggested to be improved.

Abstract:

What are the flavor characteristics and key aroma compounds of grade â…  ham? A clear description is needed.

Introduction:

In the abstract, it is stated that “for a long time, its grade has mainly been evaluated by the human nose through the three-sticks method, which is highly subjective and is not conducive to establish evaluation standards through odor markers”, but the description of “three stick method” has not been found in the introduction. Please provide additional information.

“Driven by the profits, some manufacturers use other brands of ham as partial or complete substitutes for Jinhua ham with mislabeling. This type of food fraud can hardly be identified by traditional grading method.” Is the purpose of this study to identify genuine and counterfeit products or to distinguish between different superior and inferior grades of ham? Please clarify correctly the purpose of this study.

Materials and Methods

How were the parameters set for each condition in the experiment? The result of previous experimental optimization or reference to others’ research? Please provide clarification.

Results and Discussion

The writing of this section did not meet the requirements of the “Discussion section” of the paper. A large amount of text was devoted to describing the experimental results, with little discussion and analysis. Only a full discussion and analysis could support the differences in the comparison of experimental results. Please note the correct citation of the referenced literature.

Conclusions

The odor markers of different grades Jinhua ham are what the reader most wants to see and are the most important indicators used to guide production practices. Please provide a clearer description.

References

Please check the reference styles.

Line 64: DVB/CAR/PDMS

Line 128: Check the chapter number.

Figure 1: Please display the experiment results with a bar graph.

Table1: “Ⅰ级,Ⅱ级,Ⅲ级”? Please describe in English.

Author Response

Response to Referee1

*1) Abstract:What are the flavor characteristics and key aroma compounds of grade â…  ham? A clear description is needed.

Answer: Thank you very much for your comments. I have made the changes in the summary section.“The key flavour volatiles in Grade I hams were hexanal and 2-methylbutanal. Grade I hams had strong meat aroma, pleasant fatty and roasted aroma without any off-flavours.”

*2) Introduction:

In the abstract, it is stated that “for a long time, its grade has mainly been evaluated by the human nose through the three-sticks method, which is highly subjective and is not conducive to establish evaluation standards through odor markers”, but the description of “three stick method” has not been found in the introduction. Please provide additional information.

“Driven by the profits, some manufacturers use other brands of ham as partial or complete substitutes for Jinhua ham with mislabeling. This type of food fraud can hardly be identified by traditional grading method.” Is the purpose of this study to identify genuine and counterfeit products or to distinguish between different superior and inferior grades of ham? Please clarify correctly the purpose of this study.

Answer: Thank you very much for your comments. I have added the relevant notes in the introduction section and also revised the purpose of the study.

Line75“The traditional three-sticks method is used by an experienced ham judge to determine the aroma of the sticks. The judge inserts three sticks into the ham, and the ham with a strong meat aroma and no off-flavour on any of the three sticks is judged to be Grade â… ham. This method is highly influenced by human factors.”

Line83“This experiment provided a reliable test method and evaluation basis for the rating of Jinhua ham.”

*3) Materials and Methods:How were the parameters set for each condition in the experiment? The result of previous experimental optimization or reference to others’ research? Please provide clarification.

Answer: Thank you for pointing out the deficiencies in our manuscript. I have cited references to experimental conditions in the manuscript.

Line103“We made minor adjustments according to the experimental conditions of Sa´nchez-Pen˜ a. ”

*4) Results and Discussion:The writing of this section did not meet the requirements of the “Discussion section” of the paper. A large amount of text was devoted to describing the experimental results, with little discussion and analysis. Only a full discussion and analysis could support the differences in the comparison of experimental results. Please note the correct citation of the referenced literature.

Answer: Thank you for pointing out the shortcomings in our manuscript. We have added some analysis of experimental results to the manuscript.(line204-221)

*5) Conclusions:The odor markers of different grades Jinhua ham are what the reader most wants to see and are the most important indicators used to guide production practices. Please provide a clearer description.

Answer: Thank you very much for your suggestions, we have summarised the key flavours of the three grades of ham and added to the conclusions.

*6) References:Please check the reference styles.

Answer: Thank you for pointing out the problems in our manuscript. I have revised the formatting of the references.

*7) Line 64: DVB/CAR/PDMS  

Line 128: Check the chapter number.

Figure 1: Please display the experiment results with a bar graph.

Table1: “Ⅰ级,Ⅱ级,Ⅲ级”? Please describe in English.

Answer: Thank you for pointing out the errors in the manuscript. We have corrected the errors in the manuscript and have plotted the experimental results in Figures 1-3 as bar charts.

Reviewer 2 Report

Article needs linguistic corrections in many places.

Introduction

The authors should supplement the introduction with a paragraph on the existence of other raw-maturing hams specific to the geographical region, such as Kumpiak podlaski, Pršut, Parma, etc. in addition, it should be mentioned that they differ in the breed of animals from which they are produced, the addition of spices and herbs, parameters and length of maturation, but all are dry-cured. I provide suggested literature below: https://doi.org/10.1111/ijfs.14697, https://doi.org/10.1016/j.meatsci.2019.107990, https://doi.org/10.4081/ijfs.2022.9972.

Materials and Methods

line 76- The authors should provide more details about the tested ham.

line 80- The authors should state which vials and septa with which closures they used in the experiment.

line 91-100- The authors should add the temperature of desorption of volatile compounds from the trawls in the incjector. Was it the same for all types of fibers?

line 101- chapter title is unfinished

line 109- The authors do not have standards for each volatile that is why they did a semiquantitative rather than quantitative analysis. Therefore, the designation nomenclature should be changed throughout the article.

line 120- it should be Cx, please check

Statistical Analysis- The authors should state in how many repetitions they performed each experiment.

Results and Discussion

The authors should cite figures 1 and 2 in the text where they discuss the research results presented on them. Standard deviations in figures 1, 2 and 3 are missing.

Table 1- Non-English characters are present in the table, please check; moreover, please include in the table rows with the totals quantity or % share of each group of volatile compounds.

line 169-192- Please discuss the literature regarding the presence of the identified volatile compounds in this type of ham as well as in other raw-maturing hams. You can use the previously suggested literature and other literature already cited.

Key volatiles in Jinhua ham- The authors state how some of the identified compounds smell but did not do olfactometric testing. Therefore, in order to keep the odor descriptions of each compound in the text, they must state where they got the odor descriptions from and cite these publications, databases.

The spider diagram (figure 4) is poorly readable. I suggest enlarging the perimeter or increasing its resolution. You can remove the ham image.

Conclusions

line 255- please check sentence

Conclusions are correctly formulated, containing the most important findings of the research conducted, both in terms of optimizing the analysis of volatile compounds and the differences in the odor profile of Jinhua ham grades I, II and III.

Author Response

Response to Referee2

*1) Introduction: The authors should supplement the introduction with a paragraph on the existence of other raw-maturing hams specific to the geographical region, such as Kumpiak podlaski, Pršut, Parma, etc. in addition, it should be mentioned that they differ in the breed of animals from which they are produced, the addition of spices and herbs, parameters and length of maturation, but all are dry-cured.

Answer: Thank you very much for your suggestions on the manuscript, your suggestions are valuable to us. I have added the relevant content in the introduction section.

*2) line 76- The authors should provide more details about the tested ham.

Answer: Thank you for pointing out the deficiencies in the manuscript, I have added more details about the tested hams .

Line88“Jinhua ham was selected from the hind legs of a traditional local breed of pig as the raw material. The raw material was pre-treated and then cured at 8°C for about 35 days. After curing, the ham was immersed in water at 10°C and brushed. The ham was then hung up to dry until the skin was shiny yellow and the meat was spread with oil. Finally the ham was fermented, which usually took about 5 months. Fermented and trimmed hams were stacked individually on a wooden bed according to the size and dryness of the hams, and turned over every 5-7 days. ”

*3) line 80- The authors should state which vials and septa with which closures they used in the experiment.

Answer:Thank you for pointing out the deficiencies in the manuscript. I have revised the manuscript.

Line“Chopped ham (4.5 g) was weighted and placed into 15 ml glass vials tightly capped with a PTFE septum.”

*4) line 91-100- The authors should add the temperature of desorption of volatile compounds from the trawls in the incjector. Was it the same for all types of fibers?

Answer: Thank you very much for pointing out our omission, we have added the inlet temperature and detector temperature to the manuscript. This is the same for all types of fibres. This is the same for all types of fibres.

Line114“The temperature of the inlet was set at 250°C and the temperature of the detector was set at 280°C.”

*5) line 101- chapter title is unfinished

line 109- The authors do not have standards for each volatile that is why they did a semiquantitative rather than quantitative analysis. Therefore, the designation nomenclature should be changed throughout the article.

Answer: Thank you very much for pointing out our omission, we have revised the chapters in the manuscript and added "semi-quantitative".

*6) line 120- it should be Cx, please check

Answer: Thank you very much for pointing out our errors. We have corrected the errors in the manuscript.

*7) Statistical Analysis- The authors should state in how many repetitions they performed each experiment.

Answer: Thank you very much for pointing out our omission. We repeated each experiment three times.

*8) The authors should cite figures 1 and 2 in the text where they discuss the research results presented on them. Standard deviations in figures 1, 2 and 3 are missing.

Answer: Thank you very much for your advice. We have revised Figures 1-3 and added the analysis to the manuscript.

*9) Table 1- Non-English characters are present in the table, please check; moreover, please include in the table rows with the totals quantity or % share of each group of volatile compounds.

Answer: Thank you for pointing out the problem in Table 1, we have corrected the error. And we have added the total quantity of volatiles for each group of volatile compounds to Table 1.

*10) line 169-192- Please discuss the literature regarding the presence of the identified volatile compounds in this type of ham as well as in other raw-maturing hams. You can use the previously suggested literature and other literature already cited.

Key volatiles in Jinhua ham- The authors state how some of the identified compounds smell but did not do olfactometric testing. Therefore, in order to keep the odor descriptions of each compound in the text, they must state where they got the odor descriptions from and cite these publications, databases.

Answer: Thank you very much for your comments, which have been very helpful for our manuscript. We have added more analysis to the "Results and Discussion" section. We have also added references and databases for the description of compound aromas.(line204-221)

*11) The spider diagram (figure 4) is poorly readable. I suggest enlarging the perimeter or increasing its resolution. You can remove the ham image.

line 255- please check sentence

Answer:Thank you for your comments, I have amended Figure and the sentence in line 255.

Round 2

Reviewer 2 Report

The authors have addressed all the comments made in the review. They made corrections to the tables and figures making them more informative and readable. The authors introduced recommended paragraphs in the introduction and discussion of results. Also in the methodological part, the article has been expanded with the necessary information.
I thank the authors for the corrections made, I have no further comments on the content of the manuscript.